
# Bow shock characteristics from magnetic field observations

Daniel Schmid[1] and Yasuhito Narita[1]

[1]Space Research Institute, Austrian Academy of Sciences, Schmiedlstr. 6, 8042 Graz, Austria

**Correspondence:** D. Schmid (daniel.schmid@oeaw.ac.at)

**Abstract.** Based on the magnetohydrodynamic (MHD) shock jump condition (Rankine-Hugoniot relation), we develop a tool to determine the relationship between the density jump, Alfvén Mach number and the solar wind plasma beta directly from magnetic field observations across the shock. The presented method is a useful tool to characterize the bow shock around planets, where plasma data are often limited or even not available. In particular in view of the ongoing BepiColombo mission
the tool can help to diagnose and characterize the bow shock during the planetary flybys.

## 1 Introduction

In the research field of in situ observations of the planetary magnetic and plasma environment using orbiting or flyby-passing spacecraft, one often encounters the problem that only the magnetic field data are available for detailed analyses while the plasma data (ions and electrons) are limited in many aspects such as the time resolution, the energy range, and the field-of-

view. A conventional approach is that one determines the shock normal direction from the magnetic field data by applying the coplanarity theorem, indicating that the upstream field, the downstream field, and the field jump occur in the same plane and there is no rotation of the field around the shock normal direction. By revisiting the MHD (magnetohydrodynamics) jump conditions, the so-called Rankine-Hugoniot relation, we noticed that the density jump and the Alfvén Mach number can be determined from the magnetic field data given that the adiabatic constant (or polytropic index) $\gamma$ and the plasma parameter beta

are known or set in the analysis, which we call the magnetic diagnosis of the shock condition.

## 2 Magnetic diagnosis formula

The MHD Rankine-Hugoniot relation can be reduced to a cubic algebraic equation. The solutions of the cubic equation are obtained by Zhuang and Russell (1981) using Cardano's formula and by Grabbe and Cairns (1995) using a perturbation method. In the perturbative approach, the cubic equation is solved to the first-order and the second-order accuracy for the jump variable

$X$:

$$
\begin{aligned}
X &= \rho^{(\mathrm{u})}/\rho^{(\mathrm{d})} & (1) \\
&= u_n^{(\mathrm{d})}/u_n^{(\mathrm{u})}. & (2)
\end{aligned}
$$





The jump variable represents the density jump from the downstream side to the upstream, and is equivalent to the velocity jump from the upstream side to the downstream when the mass flux conservation (continuity equation) across the shock is used,

$$\rho^{(\mathrm{u})} u_n^{(\mathrm{u})} = \rho^{(\mathrm{d})} u_n^{(\mathrm{d})} = \mathrm{const.}, \tag{3}$$

where $\rho^{(\mathrm{u})}$ is the upstream mass density, $\rho^{(\mathrm{d})}$ the downstream mass density, $u_n^{\mathrm{u}}$ the upstream flow velocity normal to the shock, and $u_n^{\mathrm{u}}$ the downstream velocity normal to the shock, In the first-order closure, the jump variable is analytically obtained (Zhuang and Russell, 1981; Grabbe and Cairns, 1995) as

$$X = \frac{\gamma - 1}{\gamma + 1} + \frac{2}{\gamma + 1}\left[\frac{1}{M_s^2} + \frac{\sin^2 \theta^{(\mathrm{u})}}{(\gamma - 1)M_\mathrm{A}^2}\right], \tag{4}$$

where $M_\mathrm{A}$ is the Alfvén Mach number, $M_s$ the sonic Mach number, $\gamma$ the adiabatic constant (assumed to be $5/3$, and $\theta^{(\mathrm{u})}$ the angle of the upstream magnetic field direction to the shock normal direction. The Alfvén Mach number and the sonic Mach number are defined, respectively, as

$$M_\mathrm{A} = \frac{u_\mathrm{n}}{V_\mathrm{A}} \tag{5}$$
$$M_\mathrm{s} = \frac{u_\mathrm{n}}{c_\mathrm{s}}. \tag{6}$$

The Alfvén Mach number refers to the Alfvén speed $V_\mathrm{A} = B/\sqrt{\mu_0 \rho}$ determined by the magnetic field magnitude $B$, the permeability of free space $\mu_0$, and the mass density $\rho$. The sonic Mach number refers to the sound speed $c_\mathrm{s} = \sqrt{\gamma p/\rho}$, which is a function of the adiabatic constant $\gamma$, the thermal pressure $p$, and the mass density.

It is useful to introduce the plasma parameter beta as

$$\beta = \frac{2\mu_0 n k_\mathrm{B} T}{B^2} = \frac{2}{\gamma}\frac{c_s^2}{V_\mathrm{A}^2}. \tag{7}$$

Equation (4) can then be arranged into

$$\frac{1}{M_\mathrm{A}^2} = \frac{\gamma + 1}{2}\left(\frac{\beta\gamma}{2} + \frac{\sin^2\theta}{\gamma - 1}\right)^{-1}\left(X - \frac{\gamma - 1}{\gamma + 1}\right), \tag{8}$$

where the sonic Mach number is expressed as a function of the plasma parameter beta and the Alfvén Mach number as

$$M_\mathrm{s}^2 = (\beta\gamma/2)^{-1}M_\mathrm{A}^2 \tag{9}$$

Equation (8) gives a constraint between the jump $X$ and the Alfvén Mach number $M_\mathrm{A}$ when the plasma beta is known, and is a valid expression in the normal incident frame (NIF, the upstream flow is normal to the shock front), the de Hoffmann-Teller frame (HT, the flow is aligned with the magnetic field), and any frame co-moving with the shock front.

We find out that Eq. (8) is conveniently solved in the HT frame. The ratio of the tangential to the normal components (or the angle from the normal direction) is the same between the flow velocity and the magnetic field in the HT frame (indicating no convective or motional electric field)

$$\frac{u_\mathrm{t}}{u_\mathrm{n}} = \frac{B_\mathrm{t}}{B_\mathrm{n}} = \tan\theta. \tag{10}$$





Equation (10) holds on the both sides of the shock. The tangential component of the momentum balance is given as

$$u_{\mathrm{t}}^{(\mathrm{d})} - u_{\mathrm{t}}^{(\mathrm{u})} = \frac{B_{\mathrm{n}}}{\mu_0 \rho u_{\mathrm{n}}} \left( B_{\mathrm{t}}^{(\mathrm{d})} - B_{\mathrm{t}}^{(\mathrm{u})} \right), \tag{11}$$

Note that the mass flux ($\rho u_{\mathrm{n}}$) and the normal component of the magnetic field ($B_{\mathrm{n}}$) are conserved quantities across the shock.

We now normalize Eq. (11) to the normal component of the flow in the downstream region, $u_n^{(\mathrm{d})}$. The left-hand side of Eq.

(11) is normalized as

$$\frac{u_{\mathrm{t}}^{(\mathrm{d})}}{u_{\mathrm{n}}^{(\mathrm{d})}} - \frac{u_{\mathrm{t}}^{(\mathrm{u})}}{u_{\mathrm{n}}^{(\mathrm{d})}} = \tan\theta^{(\mathrm{d})} - \frac{1}{X}\tan\theta^{(\mathrm{u})}, \tag{12}$$

where the jump variable $X$ is introduced with the help of Eq. (2) in the second term. The right-hand side of (11) is normalized

as

$$\frac{B_{\mathrm{n}}^2}{\mu_0 \rho u_{\mathrm{n}}} \frac{1}{u_{\mathrm{n}}^{(\mathrm{d})}} \left( \frac{B_{\mathrm{t}}^{(\mathrm{d})}}{B_{\mathrm{n}}} - \frac{B_{\mathrm{t}}^{(\mathrm{u})}}{B_{\mathrm{n}}} \right)$$

$$= \frac{\cos^2\theta^{(\mathrm{u})} B^{2(\mathrm{u})}}{\mu_0 \rho^{(\mathrm{u})} u_{\mathrm{n}}^{2(\mathrm{u})}} \frac{u_{\mathrm{n}}^{(\mathrm{u})}}{u_{\mathrm{n}}^{(\mathrm{d})}} \left( \frac{B_{\mathrm{t}}^{(\mathrm{d})}}{B_{\mathrm{n}}} - \frac{B_{\mathrm{t}}^{(\mathrm{u})}}{B_{\mathrm{n}}} \right) \tag{13}$$

$$= \frac{\cos^2\theta^{(\mathrm{u})}}{M_{\mathrm{A}}^{2(\mathrm{u})}} \frac{1}{X} (\tan\theta^{(\mathrm{d})} - \tan\theta^{(\mathrm{u})}) \tag{14}$$

where Eqs. (2) and (10) are used. By combining Eq. (12) and Eq. (14), we obtain the normalied form of Eq. (11) as

$$X\tan\theta^{(\mathrm{d})} - \tan\theta^{(\mathrm{u})} = \frac{\cos^2\theta^{(\mathrm{u})}}{M_{\mathrm{A}}^{2(\mathrm{u})}} \left( \tan\theta^{(\mathrm{d})} - \tan\theta^{(\mathrm{u})} \right). \tag{15}$$

Equations (8) and (15) form a closed set of linear equations for the jump variable $X$ and the inverse square of Alfvén Mach

number $M_{\mathrm{A}}^{-2(\mathrm{u})}$ in the upstream region, given that the plasma beta is known or set in the analysis. We arrange Eq. (8) and

Eq. (15) in a matrix form with two variables (the density jump and the Alfvén Mach number) as

$$\begin{pmatrix} X \\ M_{\mathrm{A}}^{-2} \end{pmatrix} = \begin{pmatrix} c_{11} & c_{12} \\ c_{21} & c_{22} \end{pmatrix}^{-1} \begin{pmatrix} c\frac{\gamma-1}{\gamma+1} \\ \tan\theta^{(\mathrm{u})} \end{pmatrix}, \tag{16}$$

where

$$c_{11} = c \tag{17}$$

$$c_{12} = -1 \tag{18}$$

$$c_{21} = \tan\theta^{(\mathrm{d})} \tag{19}$$

$$c_{22} = -\cos^2\theta^{(\mathrm{u})}(\tan\theta^{(\mathrm{d})} - \tan\theta^{(\mathrm{u})}). \tag{20}$$

The coefficient $c$ is

$$c = \frac{\gamma+1}{2} \left( \frac{\beta\gamma}{2} + \frac{\sin^2\theta^{(\mathrm{u})}}{\gamma-1} \right)^{-1}. \tag{21}$$





Equation (16) serves as an analysis tool to determine the jump variable and the Alfvén Mach number for the magnetic field jump across the shock under two caveats. First, the matrix in Eq. (16) undergoes a singular behavior at a certain value of beta. The singularity occurs when the determinant of the matrix vanishes, that is,

$$c_{11}\, c_{22} - c_{12}\, c_{21}$$
$$= \; -c\, \cos^2\theta^{(\mathrm{u})}(\tan\theta^{(\mathrm{d})} - \tan\theta^{(\mathrm{u})}) + \tan\theta^{(\mathrm{d})} \tag{22}$$

$$= \; 0 \tag{23}$$

In the case of singular behavior, the jump condition (Eq. 8) and the momentum balance in the HT frame (Eq. 15) are identical to each other. Second, the value of beta must be known or set. From the experimental point of view, the jump condition (Eq. 16) gives a constraint to the jump variable and the Alfvén Mach number by solving Eq. (16) in a wider range of beta.

## 3   Dependence on plasma beta and limitations of method

Figure 1 shows the solutions of Eq. (16) under a (quasi-)parallel shock configuration with an an adiabatic constant of $\gamma = 5/3$. The jump parameter, $X$, and Alfvénic Mach number, $M_{\mathrm{A}}$ are given as a function of the plasma parameter beta and various downstream shock normal magnetic field angles $\theta^{(\mathrm{d})}$ (black lines).

    In case of a (quasi-)parallel shock ($\theta^{(\mathrm{u})} \approx 1°$), the determinant in Eq.(22) vanishes for $\beta \approx 1.2$ and $\theta^{(\mathrm{d})} \approx 4°$ and the matrix in Eq. (16) becomes singular (dashed-dotted line). For $\beta > 1.2$ the jump parameter $X$ and Alfvénic Mach number $M_{\mathrm{A}}$ increases

with the plasma beta and/or increasing downstream shock normal angle until $\theta^{(\mathrm{d})} \approx 4°$ is reached. The lower limit of the plasma beta shows the limitation of the first-order solutions of Eq. (4). A physical explaination for this lower limit might be that in the special case of parallel MHD shocks the magnetic field does not affect the shock and the bow shock becomes identical to a hydrodynamic shock of neutral fluids. In such a case the shock is entirely characterized by the sonic Mach number $M_{\mathrm{s}}$ and can only emerge when $M_{\mathrm{s}}^2 > 1$. Following Eq. (9), this implies that $\beta\gamma/2 > 1$ ($\beta > 1.2$ for $\gamma = 5/3$) needs to be fullfilled to create

a fast magnetosonic shock. With increasing upstream shock normal angle the Alfvénic Mach number becomes the dominating parameter for the shock formation and the limit of the plasma beta is shifted towards lower plasma betas, and subsequently vanishes for $\theta^{(\mathrm{u})} \geq 35°$. It should be mentioned that Eq. (16) also yields results for plasma betas below the lower limit (here $\beta < 1.2$). However, these solutions are not physically meaningful for planetary bow shocks.

    In practice $\theta^{\mathrm{d}}$ is directly given by the downstream magnetic field measurements. By assuming a reasonable range for the

solar wind plasma beta it is therfore possible to approximate the jump parameter $X$ and the Alfvénic Mach number $M_{\mathrm{A}}$ directly from Eq. (16).

## 4   Summary and outlook

Here we present a useful tool to diagnose the bow shock condition around planets on basis of magnetic field observations. From the upstream and downstream shock normal angle of the magnetic field, it is possible to approximate the relation between



**Figure 1.** Alfvénic Mach number, $M_A$, and jump parameter, $X$, as a function of the solar wind plasma beta under various shock normal angles of the downstream magnetic field $\theta^{(d)}$ (solid lines) in case of a (quasi-)parallel shock configurations ($\theta^{(u)} = 1°$). The dashed-dotted line illustrates the plasma beta and downstream shock normal angle where the matrix in Eq. (16) becomes singular.



compression ratio, Alfvénic Mach number and the solar wind plasma beta. The presented work also shows the limitation of the first-order approximation of the jump variable. In a future work the higher-order solutions should be taken into account.

The tool is particularly helpful to study the solar wind conditions and bow shock characteristics during the planetary flybys of the ongoing BepiColombo mission, as the plasma data are quite limited due to the restricted field-of-view of the plasma instruments during the cruise phase. Gedalin et al., (2022) developed a method to determine the Mach number of the low-Mach

number shock based on magnetic field data only. With the method presented in this paper, it is therefore possible to get an approximate value of the plasma parameter beta.

*Code and data availability.*  No codes or data are used in this work.

*Author contributions.*  All authors listed have made a substantial, direct, and intellectual contribution to the work and approved it for publication.

*Competing interests.*  No conflict of Interest.

*Acknowledgements.*  The authors thank Prof. Iver Cairns for encouraging the study.





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
