# Peer review of "Bow shock characteristics from magnetic field observations"

_Annales Geophysicae, 2022_

## Referee Comment (RC1)

**Review on angeo-2022-30**

The paper proposes a method to derive the bow shock characteristics from magnetic field measurements only. Unfortunately, the proposed method does not work. Rankine-Hugoniot relations give the function

$$R = R(M, \theta, \beta) \tag{1}$$

where $R = B_d/B_u$ and the rest of the parameters are defined in the paper. Inverting, one has

$$M = M(R, \theta, \beta) \tag{2}$$

Assuming the both $R$ and $\theta$ can be obtained from the magnetic field measurements only, there is still dependence on $\beta$, which requires particle measurements. This dependence could be ignored if weak. Figure 1 upper panel shows that this assumption is not correct. Even more important, Figure 1 shows that the Mach number is extremely sensitive to the value of the shock angle. The latter is not measured with the precision of $0.5°$.

---

## Author Comment (AC1)

We thank the referee for his/her careful reading and raising thoughtful comments.

> The paper proposes a method to derive the bow shock characteristics from
> magnetic field measurements only. Unfortunately, the proposed method does
> not work. Rankine-Hugoniot relations give the function
> R = R(M, θ, β) (1)
> where R = Bd/Bu and the rest of the parameters are defined in the paper.
> Inverting, one has
> M = M (R, θ, β) (2)
> Assuming the both R and θ can be obtained from the magnetic field measure-
> ments only, there is still dependence on β, which requires particle measurements.

Reply

Referee's view with his/her Eqs. (1) and (2) are correct. It is true that the plasma beta needs to be specified to derive the Mach number as in his/her Eq. (2), but it is not correct that particle measurements are required. Particle measurements are the ideal input to our method if any. Even if no particle data are available, Eq. (2) above gives us a useful relation between the measured magnetic field jump (R and theta) and the Mach number (M) parametrized by the plasma beta. This relation serves as a constraint of estimating the Mach number. In other words, one obtains a range of Mach number for a given range of plasma beta. Alone this property advances the interpretation of the magnetic field data significantly. The treatment of the missing beta information is elaborated in the revision.

> This dependence could be ignored if weak. Figure 1 upper panel shows that this
> assumption is not correct. Even more important, Figure 1 shows that the Mach
> number is extremely sensitive to the value of the shock angle. The latter is not
> measured with the precision of 0.5◦.

Reply

The dependence is a good point. We have the dependence issue with respect to the shock angle (between the upstream magnetic field and the shock normal) and the plasma beta. Our method is insensitive to the shock angle and the plasma beta for a quasi-perpendicular shock crossing (Figure to be added in the revision). Our method is sensitive to these parameters (shock angle and beta) for a quasi-parallel shock crossing, since the analysis (matrix inversion) is made near the singularity in the parameter space. We have shown the quasi-parallel shock crossing to warn or emphasize the problem of singularity, but we should compare between the quasi-perpendicular shock case (a safe case) and the quasi-parallel shock case (a risky case), which will be done in the revision.

Other changes planned

The title with "bow shock" is misleading and will be corrected into "magnetohydrodynamic shock" in the revision.